# Blood metal levels and serum testosterone concentrations in male and female children and adolescents: NHANES 2011–2012

Qi Yao[1]©, Ge Zhou[2]©, Meilin Xu[3], Jianguo Dai[1], Ziwei Qian[1], Zijing Cai[1], Luyao Zhang[1], Yong Tan[2]*, Rongkui Hu®[2]*

**1** Department of Pathology and Pathophysiology, Nanjing University of Chinese Medicine, Nanjing, Jiangsu Province, China, **2** Department of Reproductive Medicine, Affiliated Hospital of Nanjing University of Chinese Medicine, Jiangsu Province Hospital of Chinese Medicine, Nanjing, Jiangsu Province, China, **3** Medical department life science China, GE healthcare China, Beijing, China

© These authors contributed equally to this work.
* xiangyu198110@163.com (RH); xijun1025@163.com (YT)

**Data Availability Statement:** All relevant data are within its Supporting Information files.

**Funding:** This work was supported by grants from the National Natural Science Fund for Young

## Abstract

Environmental exposure to metals is ubiquitous, but its relation to androgen hormone levels is not well understood, especially in children and adolescents. This study aimed to explore the relationship between blood metal concentrations (lead, cadmium, total mercury, selenium, and manganese) and serum total testosterone (TT) levels in 6–19-year-old children and adolescents in the National Health and Nutrition Examination Survey (NHANES) 2011–2012. Weighted multivariable linear regression models using NHANES sampling weights were employed to evaluate the association between log-transformed serum TT and each metal categories in male and female children (age 6-11years) and adolescents (age 12–19 years). We established that blood cadmium and manganese levels were associated with significantly higher serum TT levels in the female adolescents. Additionally, the blood selenium levels in male adolescents were related to significantly higher serum TT. No significant associations between blood lead or total mercury levels and TT were observed in children or adolescents of either sex. These findings suggest that environmental exposure to certain metals could affect serum TT levels in adolescents, which might have important implications for the health of adolescents. Further research is required to confirm and extend our present findings.

## Introduction

Testosterone is a principal sex hormone needed for the normal physiologic processes during all life stages. In males, testosterone is essential for the development and maintenance of secondary sexual traits [1–2]. Testosterone also influences bone mass, muscle strength, mood, and intellectual capacity [1–2]. In females, testosterone is of crucial importance for bone density and is necessary for normal ovarian and sexual function, libido, energy, and cardiovascular and cognitive functions [3–4].

Scholars (Grant No. 81704164), Natural Science Foundation of the Jiangsu Province (Grant No. BK20151043), Six Talent Peaks Project in the Jiangsu Province (Grant No. WSN-044), and 333 Talent Project in the Jiangsu Province (Grant No. 2016III-3288). The funders had no role in study design, data collection and analysis, decision to publish, or preparation of the manuscript.

**Competing interests:** The authors have declared that no competing interests exist.

Testosterone imbalances lead to reproductive dysfunction during multiple life stages in both sexes. Low testosterone levels are related to reduced semen quality in men [5, 6], increased genital malformations [7], and changes in the time of onset and/or progression of puberty [8–9]. On the other hand, high testosterone levels are linked to polycystic ovary syndrome (PCOS) in females [10], increased genital system cancers [11, 12], and altered pubertal development [13, 14]. Therefore, research on the factors affecting testosterone levels in both sexes is needed.

Metals have been shown to interact with testosterone levels. The general population is exposed to low concentrations of metals daily through the consumption of water, foods, supplements, and inhalation of air. Due to widespread metal exposure of humans, a growing concern exists regarding their endocrine-disrupting potential. Non-essential metal elements, such as cadmium, lead, and mercury, have endocrine-disrupting properties even at low concentrations. A positive relationship between environmental cadmium exposure and testosterone levels has also been shown among adult males and females [15–20], and several epidemiologic studies have indicated that environmental lead exposure is associated with increased testosterone levels in U.S., Croatian, and Chinese men [21–23].The research regarding environmental mercury exposure and testosterone levels in humans is limited. A study by Barregârd et al. showed that serum total testosterone (TT) concentrations were positively correlated with cumulative mercury exposure [24]. Essential metals, such as manganese and selenium, can be harmful depending on their concentrations. Rats exposed to low manganese levels had increased testosterone levels, while high manganese levels resulted in decreased testosterone levels [25–27]. Exposing adult male rats to low selenium concentrations resulted in increased testosterone levels, while high levels resulted in decreased testosterone levels [28, 29].

In summary, most studies assessing the reproductive endocrine-disrupting effect of low, environmentally-relevant metal exposures in adults have focused on non-essential metals (cadmium and lead), whereas information on other potential endocrine-disrupting metals is still scant. Moreover, children and adolescents are at the ages where susceptibility to the adverse health effects of endocrine disruptors is most concerning. Few studies have focused on the relationship between environmental metal exposure and endogenous androgen hormone levels in children and adolescents. Therefore, our objective was to examine the potential associations between blood metal (cadmium, lead, mercury, manganese, and selenium) and circulating serum TT levels in a general population sampling of 6–11-year-old children and 12–19-year-old adolescents in the United States.

## Materials and methods

### Study population

We analyzed the data from the National Health and Nutrition Examination Survey (NHANES) 2011–2012. NHANES is a cross-sectional, U.S.-representative survey conducted annually by the Centers for Disease Control and Prevention (CDC). The goal of NHANES is to assess the health and nutritional status of the general U.S. population. The data were collected by questionnaire surveys, household interviews, physical examinations, and laboratory tests. We analyzed data from a subset of male children (age 6–11 years), male adolescents (age 12–19 years), female children (age 6–11 years), and female adolescents (age 12–19 years). Participants with missing data on blood metal levels, serum TT, and covariates were excluded from the analysis. In the final sample size, we assessed data from 431 male children, 493 male adolescents, 426 female children, and 470 female adolescents. NHANES received approval from the National Center for Health Statistics (NCHS) Ethics Review Board, and informed consent was obtained for all participants.

## 2.2 Blood sample

Whole venous blood were collected from participants at the Mobile Examination Center (MEC), which were then processed, stored, and shipped to laboratories at the CDC (Atlanta, GA, USA) for analysis. Samples were analyzed for blood metals, serum TT and serum cotinine.

## 2.3 Serum total testosterone levels

Serum TT levels were analyzed by isotope-dilution liquid chromatography-tandem mass spectrometry. Information regarding the reliability, validation, and quality control of serum TT levels is presented in detail in the NHANES laboratory methods (http://www.cdc.gov/nchs/data/nhanes/nhanes_11_12/TST_G_met.pdf).

## 2.4 Blood metals

Whole blood lead, cadmium, total mercury, manganese, and selenium concentrations were determined using inductively coupled plasma mass spectrometry-based on quadrupole ICP-MS technology. The methodologic details of blood metal detection and measurements are described in the NHANES laboratory methods(http://www.cdc.gov/nchs/data/nhanes/nhanes_11_12/PbCd_G_met_blood%20metals.pdf). The detection limit for all analytes was constant in the data set. The lower detection limit(LOD) for lead was 0.25 μg/dL, whereas it was 0.16 μg/L for cadmium, 30 μg/L for selenium, 1.06 μg/L for manganese, and 0.16 μg/L for total mercury. Values below the LOD were imputed by LOD and divided by the square root of 2, which provided an imputed value for the individuals who had levels below the LOD.

## 2.5 Covariates

We examined the following as potential confounding variables: age, race/ethnicity, poverty income ratio (PIR), obesity, seasons of collection, times of venipuncture, and serum cotinine as a biomarker of exposure to environmental tobacco smoke. The selection of these confounders was based on literatures [17, 30].Race/ethnicity was categorized as Non-Hispanic White, Non-Hispanic Black, Hispanic (Mexican American and other Hispanic black), and others. PIR represents the calculated family income to the poverty ratio threshold. Children and adolescents were classified as normal/underweight, overweight, or obese according to their age and sex, in compliance with the criteria defined by NHANES (Body Measures File; https://wwwn.cdc.gov/Nchs/Nhanes/2011-2012/BMX_G.htm). The collection season was obtained from the NHANES demographic data pertaining to the six-month period when the examinations were conducted, which was then classified into two categories, including November 1st–April 30th and May 1st–October 31st. The time of venipuncture, which can be found in the NHANES Fasting Questionnaire File, was categorized as morning, afternoon, or evening sessions. Serum cotinine was measured with isotope dilution-high performance liquid chromatography/atmospheric pressure chemical ionization tandem mass spectrometry (ID HPLC-APCI MS/MS). The methodological details of the detection and measurements of serum cotinine the levels are described in the NHANES laboratory methods (http://www.cdc.gov/nchs/data/nhanes/nhanes_11_12/COT_G_met_cotinine.pdf).

## 2.6. Statistical analysis

All statistical analyses were performed using Empower (R) (www.empowerstats.com, X&Ysolutions, inc.Boston, MA) and R (http://www.R-project.org) software. Descriptive statistics of participant demographics and concentrations of the metals was calculated. Lead, total mercury, manganese, and selenium were categorized into quartiles because more than 75% of the samples

were above LOD. The grouping cut-point for cadmium was determined by the percentage of samples LOD. The low group consisted of all subjects with blood cadmium <LOD, whereas the medium and high groups consisted of equal-sized bins among the detected values. Serum TT was log-transformed for analyses, because the distribution of serum TT was skewed left. Serum cotinine was log-transformed as well. We employed weighted multivariable linear regression models using NHANES sampling weights to evaluate the association between log-transformed serum TT and each metal categories adjusted for the covariates described in section 2.5. Model 1 controlled for age, race and BMI. Model 2 controlled for PIR, seasons of collection, times of venipuncture, and serum cotinine, in addition to the covariates of model 1. Because our dependent variable, serum TT, was log-transformed, the estimated coefficients (β) in the weighted regression models were transformed back by exponentiation of β (exp(β)) and presented as percent differences (equation: % change = [exp(β)-1]× 100). Since the analyses were performed for each subgroup (male children, male adolescents, female children, and female adolescents) separately, the *p*-values were adjusted for multiple testing using least significant difference (LSD) method. *P*-values less than 0.05 were considered to be statistically significant.

## Results

The characteristics of the study population by age and sex categories are presented in Table 1. The median serum TT levels of male children, male adolescents, female children, and female adolescents were 3.16 ng/dL, 354.67 ng/dL, 4.86 ng/dL, and 23.95 ng/dL, respectively.

**Table 1. Characteristics of the 6–19-year-old children and adolescent participants in NHANES 2011–2012.**

| Parameter | Male children | Male adolescents | Female children | Female adolescents |
|---|---|---|---|---|
| n | 431 | 493 | 426 | 470 |
| Age (years) | 9 (7–10) | 15(14–17) | 9 (7–10) | 15(13–17) |
| Serum total testosterone (ng/dL) | 3.16 (1.80–5.68) | 354.67 (208.08–496.58) | 4.86 (2.69–10.29) | 23.95 (16.72–31.55) |
| Serum cotinine (ng/mL) | 0.035 (0.011–0.242) | 0.038 (0.011–0.431) | 0.039 (0.011–0.181) | 0.028 (0.011–0.210) |
| Ratio family income to poverty | 1.26(0.74–2.70) | 1.57 (0.86–3.40) | 1.30 (0.70–2.78) | 1.40 (0.69–2.96) |
| Obesity[a] | | | | |
| Normal/underweight | 263 (61.02%) | 309 (62.68%) | 266 (62.44%) | 297 (63.19%) |
| Overweight | 66 (15.31%) | 77 (15.62%) | 68 (15.96%) | 76 (16.17%) |
| Obese | 102 (23.67%) | 107 (21.70%) | 92 (21.60%) | 97 (20.64%) |
| Race/ethnicity | | | | |
| Non-Hispanic White | 113 (26.22%) | 119 (24.14%) | 98(23.01%) | 105 (22.34%) |
| Non-Hispanic black | 116 (26.91%) | 154 (31.24%) | 128(30.05%) | 141 (30.00%) |
| Hispanic | 145 (33.64%) | 138 (27.99%) | 129(30.28%) | 144 (30.64%) |
| Other | 57 (13.23%) | 82 (16.63%) | 71(16.67%) | 80 (17.02%) |
| Session time of venipuncture | | | | |
| Morning | 177 (41.07%) | 243 (49.29%) | 207 (48.59%) | 237 (50.43%) |
| Afternoon | 157 (36.43%) | 170 (34.48%) | 149 (34.98%) | 156 (33.19%) |
| Evening | 97 (22.51%) | 80 (16.23%) | 70 (16.43%) | 77 (16.38%) |
| Six-month period when the examination was performed | | | | |
| 1 November through 30 April | 216 (50.12%) | 237 (48.29%) | 221 (51.88%) | 216 (45.96%) |
| 1 May through 31 October | 215 (49.88%) | 256 (51.93%) | 205 (48.12%) | 254 (54.04%) |

Data are summarized as median (interquartile range) for continuous variables or as number with proportion for categorical variables.
[a]Children and adolescents were classified as normal/underweight, overweight, or obese according to their age and sex, as defined by NHANES (http:/wwwn.cdc.gov/Nchs/Nhanes/2011-2012/BMX_G.htm)

**Table 2. The distribution of metal concentrations in the blood of the 6–19-year-old children and adolescent participants in NHANES 2011–2012.**

| Parameter | Male children | | | | Male adolescents | | | |
|---|---|---|---|---|---|---|---|---|
| n | 431 | | | | 493 | | | |
| | N(%)<LOD | Geometric | median | interquartile | N(%)<LOD | Geometric | median | interquartile |
| Blood lead (µg/dL) | 7 (1.6%) | 0.76 | 0.72 | 0.52–1.02 | 11 (2.2%) | 0.68 | 0.66 | 0.47–0.96 |
| Blood cadmium (µg/L) | 311 (72.2%) | 0.13 | 0.11 | 0.11–0.16 | 237 (48.1%) | 0.17 | 0.16 | 0.11–0.23 |
| Blood mercury, total (µg/L) | 69 (16.0%) | 0.35 | 0.34 | 0.20–0.59 | 48 (9.7%) | 0.47 | 0.43 | 0.23–0.79 |
| Blood selenium (µg/L) | 0 (0.0%) | 174 | 175 | 163–188 | 0 (0.0%) | 187 | 187 | 174–202 |
| Blood manganese (µg/L) | 0 (0.0%) | 9.79 | 9.72 | 7.86–11.84 | 0 (0.0%) | 9.43 | 9.33 | 7.62–11.54 |
| Parameter | Female children | | | | Female adolescents | | | |
| n | 426 | | | | 470 | | | |
| | N(%)<LOD | Geometric | median | interquartile | N(%)<LOD | Geometric | median | interquartile |
| Blood lead (µg/L) | 6 (1.4%) | 0.68 | 0.65 | 0.48–0.93 | 27 (5.8%) | 0.47 | 0.47 | 0.35–0.63 |
| Blood cadmium (µg/L) | 277 (65.0%) | 0.14 | 0.11 | 0.11–0.18 | 194 (41.3%) | 0.19 | 0.18 | 0.11–0.27 |
| Blood mercury, total (µg/L) | 61 (14.3%) | 0.39 | 0.36 | 0.22–0.66 | 40 (8.5%) | 0.50 | 0.49 | 0.26–0.83 |
| Blood selenium (µg/L) | 0 (0.0%) | 177 | 177 | 165–189 | 0 (0.0%) | 184 | 183 | 169–200 |
| Blood manganese (µg/L) | 0 (0.0%) | 10.55 | 10.52 | 8.72–12.90 | 0 (0.0%) | 10.75 | 10.84 | 8.64–13.26 |

Abbreviations: LOD, limit of detection

As can be seen in Table 2, which shows the distribution of blood metals concentrations by age and sex categories, blood manganese and selenium levels were >LOD for all samples. >75% of the samples had blood lead and total mercury levels >LOD in children and adolescents. 27.8% of the samples had blood cadmium levels >LOD in male children.35.0% of the samples had blood cadmium levels >LOD in female children.51.9% of the samples had blood cadmium levels >LOD in male adolescents.58.7%% of the samples had blood cadmium levels >LOD in female adolescents.

In female adolescents, serum TT was significantly higher for girls in the 4th *vs.* 1st quartile of lead exposure, but there were no significant trends with increasing quartiles of exposure (adjusted *p*-trend = 0.14, based on model 1). Blood lead levels did not appear to be associated with serum TT concentrations in male and female children and male adolescents, and consistent trends with increasing quartiles of exposure were not evident (Table 3).

As can be observed in Table 4, serum TT concentrations were positively associated with blood cadmium in female adolescents, and a monotonic increase was seen in the mean serum TT concentration with increasing exposure (adjusted *p*-trend = 0.004, based on model 1). Significantly higher serum TT levels for the 2nd and 3rd blood cadmium exposure levels (13.24%; 95% CI:3.41%, 24.00%, and 23.15%; 95% CI:6.43%, 42.5%, respectively, based on model 1) compared with the lowest exposure levels were also seen. This pattern of association persisted following an adjustment for age, BMI, race, serum cotinine, time of venipuncture, collection season, family income to poverty ratio, and the *p*-trend remained significant (adjusted *p*-trend = 0.002, based on model 2). In male adolescents, serum TT levels were significantly higher in boys in the 3rd *vs.*1st levels of lead exposure, and the trending *p*-value was significant (adjusted *p*-trend = 0.04, based on model1). However, the associations were not significant after adjustment for age, BMI, race, serum cotinine, time of venipuncture, collection season, family income to poverty ratio (adjusted *p*-trend = 0.62, based on model 2). Blood cadmium did not appear to be associated with serum TT concentrations in male and female children, and consistent trends with increasing exposure were not evident.

**Table 3. Percent differences (95% CI) in the serum TT by quartiles of blood lead exposure, NHANES, 2011–2012 using weighted regression models.**

| Blood lead(µg/dL) | Model 1[a] | Model2[b] |
|---|---|---|
| Male children(N = 431) | | |
| ≤0.52(n = 110) | Reference | Reference |
| 0.52–0.72(n = 111) | 4.1 (-18.47, 32.9) | 11.75 (-13.06,43.65) |
| 0.72–1.02 (n = 105) | -6.13(-27.64, 21.77) | -4.63(-26.97, 24.55) |
| >1.02 (n = 105) | -12.83 (-33.68, 14.58) | -13.09 (-34.45,15.22) |
| Adjusted *p*-trend | 0.36 | 0.42 |
| Male adolescents(N = 493) | | |
| ≤0.47(n = 129) | Reference | Reference |
| 0.47–0.66 (n = 120) | -3.36 (-20.98,18.2) | -4.35 (-21.22,16.14) |
| 0.66–0.96 (n = 121) | 14.99 (-7.77,43.37) | 8.15 (-12.91,34.3) |
| >0.96 (n = 123) | 15.62 (-7.07,43.86) | 6.32 (-14.62,32.4) |
| Adjusted *p*-trend | 0.18 | 0.58 |
| Female children(N = 426) | | |
| ≤0.48(n = 109) | Reference | Reference |
| 0.48–0.65 (n = 106) | 14.34 (-3.75,35.81) | 14.9 (-3.54,36.86) |
| 0.65–0.93 (n = 106) | -5.00 (-21.05,14.32) | -0.96 (-17.80,19.34) |
| >0.93 (n = 105) | -5.73 (-23.13,15.61) | -2.40 (-21.00,20.57) |
| Adjusted *p*-trend | 0.36 | 0.63 |
| Female adolescents(N = 470) | | |
| ≤0.35 (n = 122) | Reference | Reference |
| 0.35–0.47 (n = 118) | -8.55 (-18.52,2.63) | -7.83 (-18.22,3.88) |
| 0.47–0.63 (n = 113) | -1.95 (-13.04,10.56) | -1.07 (-12.67,12.06) |
| >0.63 (n = 117) | 13.12 (0.06,27.88) | 14.85 (0.83,30.81) |
| Adjusted *p*-trend[c] | 0.14 | 0.08 |

[a]Adjusted for age (continuous), BMI (normal/underweight, overweight, and obese) and race-ethnicity (Non-Hispanic White, Non-Hispanic black, Hispanic, and other).

[b] Adjusted for variables in model 1 plus serum cotinine (log-transformed, continuous), time of venipuncture (morning, afternoon, and evening), season of collection (1 November through 30 April, 1 May through 31 October), ratio family income to poverty (continuous).

c Adjusted *p*-trend:*p*-values adjusted for multiple testing.

In female children, serum TT was significantly higher for girls in the 4th*vs.*1st quartile of mercury exposure, but there were no signficant trends with increasing quartiles of exposure (adjusted *p*-trend = 0.10, based on model 1). Blood mercury did not appear to be associated with serum TT in male and female adolescent and male children, and no evidence of consistent trends with increasing quartiles of exposure was established(Table 5).

In male adolescent subjects, the mean serum TT level was higher for boys in the 3rd and 4th quartiles of blood selenium versus those in the 1st quartile, and the trend *p*-value was significant (adjusted *p*-trend = 0.0002, based on model 1) (Table 6). However, the quartile-specific increase was significant only for the 4th quartile (43.27%; 95% CI:14.20%, 79.73%, based on model 1). The association were generally consistent with model1 after adjustment for age, BMI, race, serum cotinine, time of venipuncture, season of collection, ratio family income to poverty, and the *p*-trend remained significant(adjusted *p*-trend = 0.003 based on model 2). Blood selenium did not appear to be associated with the serum TT level in male and female

**Table 4. Percent differences (95% CI) in serum TT betweenthree blood cadmium exposure levels, NHANES, 2011–2012using weighted regression models.**

| Blood cadmium(μg/L) | Model 1[a] | Model2[b] |
|---|---|---|
| Male children(N = 431) | | |
| ≤0.11 (n = 311) | Reference | Reference |
| 0.11–0.20 (n = 73) | 6.87 (-16.33,36.51) | 17.65 (-8.02,50.47) |
| >0.20 (n = 47) | -14.39 (-39.09,20.3) | -8.04 (-34.62,29.35) |
| Adjusted p-trend | 0.86 | 0.83 |
| Male adolescents(N = 493) | | |
| ≤0.11 (n = 237) | Reference | Reference |
| 0.11–0.23 (n = 206) | 11.31 (-5.45,31.05) | 3.56 (-11.68,21.43) |
| >0.23 (n = 50) | 37.97 (4.45,82.26) | 21.68 (-11.70,67.68) |
| Adjusted p-trend | 0.04 | 0.62 |
| Female children(N = 426) | | |
| ≤0.11 (n = 227) | Reference | Reference |
| 0.11–0.20 (n = 91) | 16.19 (-1.16,36.59) | 18.24 (0.6,38.98) |
| >0.20 (n = 58) | -7.95 (-25.67,13.99) | -8.64 (-26.22,13.13) |
| Adjusted p-trend | 0.86 | 0.83 |
| Female adolescent(N = 470) | | |
| ≤0.11 (n = 194) | Reference | Reference |
| 0.11–0.25 (n = 200) | 13.24 (3.41,24) | 14.28 (4.31,25.19) |
| >0.25 (n = 76) | 23.15 (6.43,42.5) | 30.79 (9.7,55.94) |
| Adjusted p-trend[c] | 0.0038 | 0.0015 |

[a] Adjusted for age (continuous), BMI (normal/underweight, overweight, and obese) and race-ethnicity (Non-Hispanic White, Non-Hispanic black, Hispanic and other).

[b] Adjusted for variables in model 1 plus serum cotinine (log-transformed, continuous), time of venipuncture (morning, afternoon, and evening), season of collection (1 November through 30 April, 1 May through 31 October), ratio family income to poverty.

c Adjusted p-trend:p-values adjusted for multiple testing.

children and female adolescents, and no evidence of consistent trends with increasing quartiles of exposure was established.

As seen in Table 7, according to model 1, TT was significantly higher in all quartiles of blood manganese than in the lowest quartile in all population subgroups of female adolescents. The trend p-value was significant for female adolescents (adjusted p-trend = 0.001, based on model 1). After adjustment for age, the values of BMI, race, serum cotinine, time of venipuncture, season of collection, ratio of family income to poverty, and the p-trend remained significant (adjusted p-trend = 0.0008, based on model 2). No significant associations were found between serum TT and blood manganese levels in male and female children and male adolescents, and no evidence of consistent trends with increasing quartiles exposure was established.

## Discussion

The aim of this study was to investigate the associations between blood metal (lead, cadmium, total mercury, manganese, and selenium) and serum TT levels in male and female children and adolescents. In the present cross-sectional analysis of data from NHANES 2011–2012, blood cadmium and manganese levels were positively associated with serum TT levels in female adolescents. Additionally, blood selenium was positively associated with serum TT

**Table 5. Percent differences (95% CI) in serum TT by quartiles of blood mercury exposure, NHANES, 2011–2012using weighted regression models.**

| Blood mercury(μg/L) | Model 1[a] | Model2[b] |
|---|---|---|
| Male children(N = 431) | | |
| ≤0.20 (n = 112) | Reference | Reference |
| 0.20–0.34 (n = 105) | 2.45 (-20.36,31.79) | 5.49 (-17.59,35.03) |
| 0.34–0.59 (n = 111) | 15.02 (-10.17,47.28) | 10.29 (-13.62,40.82) |
| >0.59 (n = 103) | 4.92 (-21.02,39.40) | 0.67 (-24.36,33.99) |
| Adjusted *p*-trend | 0.46 | 0.75 |
| Male adolescents(N = 493) | | |
| ≤0.23 (n = 125) | Reference | Reference |
| 0.23–0.43 (n = 123) | 18.47 (-3.05,44.76) | 13.75 (-6.45,38.32) |
| 0.43–0.79 (n = 123) | -10.63 (-28.14,11.14) | -8.68 (-26.14,12.91) |
| >0.79 (n = 122) | 21.77 (-2.77,52.5) | 19.28 (-4.21,48.55) |
| Adjusted*p*-trend | 0.46 | 0.70 |
| Female children(N = 426) | | |
| ≤0.22(n = 114) | Reference | Reference |
| 0.22–0.36 (n = 103) | 6.76 (-10.22,26.94) | 8.62 (-9.17,29.89) |
| 0.36–0.66 (n = 106) | 0.89 (-15.07,19.83) | 0.7 (-15.56,20.09) |
| >0.66 (n = 103) | 29.78 (7.96,56.00) | 29.15 (6.9,56.02) |
| Adjusted*p*-trend | 0.10 | 0.14 |
| Female adolescent(N = 470) | | |
| ≤0.26 (n = 120) | Reference | Reference |
| 0.26–0.49 (n = 118) | -4.72 (-14.67,6.39) | -2.75 (-13.25,9.03) |
| 0.49–0.83 (n = 116) | -2.45 (-13.49,10.00) | 0.05 (-11.56,13.17) |
| >0.83 (n = 116) | -8.28 (-19.6,4.64) | -5.84 (-17.78,7.83) |
| Adjusted*p*-trend[c] | 0.46 | 0.70 |

[a] Adjusted for age (continuous), BMI(normal/underweight, overweight, and obese)and race-ethnicity(Non-Hispanic White, Non-Hispanic black, Hispanic and other).

[b] Adjusted for variables in model 1 plus serum cotinine (log-transformed, continuous), time of venipuncture (morning, afternoon and evening), season of collection(1 November through 30 April, 1 May through 31 October), ratio family income to poverty (continuous).

c Adjusted *p*-trend:*p*-values adjusted for multiple testing.

levels in male adolescents. No significant associations were observed in the children (male or female).

Circulating testosterone in the body includes sex hormone-binding globulin (SHGB)-bound testosterone, albumin-bound testosterone, corticosteroid-binding globulin (CBG)-bound testosterone, and in an unbound or free form. SHBG-bound testosterone is tightly bound and unavailable to cells. Albumin-bound testosterone and CGB-bound testosterone are weekly bound and dissociate from testosterone rapidly [31]. The term bioavailable testosterone refers to the sum of the albumin-bound, CGB-bound, and free components, which represents the testosterone fraction available to cells. In the present study, we evaluated serum TT, which included both the bioavailable and SHGB-bound forms.

Lead and cadmium, as well-known reproductive toxicants and endocrine disruptor compounds, can cause Leydig cell damage and/or hormone imbalances. *In vivo* and *in vitro* studies showed that lead and cadmium might directly affect Leydig cell function by impairing steroidogenesis[32–36].Lead and cadmium also might disrupt hypothalamic-pituitary-testicular

**Table 6. Percent differences (95% CI) in serum TT by quartiles of blood selenium exposure, NHANES, 2011–2012 using weighted regression models.**

| Blood selenium(µg/L) | Model 1[a] | Model2[b] |
|---|---|---|
| Male children(N = 431) | | |
| ≤163 (n = 108) | Reference | Reference |
| 163–175 (n = 109) | -4.76 (-27.11,24.45) | -2.05 (-24.7,27.41) |
| 175–188 (n = 107) | -16.69 (-36.08,8.58) | -13.07 (-32.96,12.71) |
| >188 (n = 107) | -11.75 (-32.55,15.46) | -8.28 (-29.53,19.37) |
| Adjusted *p*-trend | 0.32 | 0.37 |
| Male adolescents(N = 493) | | |
| ≤174 (n = 126) | Reference | Reference |
| 174–187 (n = 121) | -17.49 (-34.47,3.91) | -18.3 (-34.67,2.16) |
| 187–202 (n = 123) | 2.33 (-18.44,28.39) | -4.98 (-23.91,18.67) |
| >202 (n = 123) | 43.27 (14.2,79.73) | 33.38 (6.99,66.29) |
| Adjusted *p*-trend | 0.0002 | 0.0027 |
| Female children(N = 426) | | |
| ≤165 (n = 107) | Reference | Reference |
| 165–177 (n = 106) | -16.76 (-30.82,0.18) | -17.64 (-31.6,-0.83) |
| 177–189 (n = 107) | -17.62 (-31.59,-0.80) | -21.52 (-35.01,-5.22) |
| >189 (n = 106) | -8.78 (-23.63,8.96) | -11.93 (-26.47,5.49) |
| Adjusted *p*-trend | 0.36 | 0.23 |
| Female adolescents(N = 470) | | |
| ≤169 (n = 118) | Reference | Reference |
| 169–183 (n = 117) | 4.68 (-7.33,18.25) | 4.55 (-7.51,18.19) |
| 183–200 (n = 118) | -6.45 (-16.98,5.41) | -7.79 (-18.3,4.08) |
| >200 (n = 117) | -7.93 (-18.29,3.75) | -8.23 (-18.56,3.42) |
| Adjusted *p*-trend[c] | 0.13 | 0.10 |

[a] Adjusted for age (continuous), BMI (normal/underweight, overweight, and obese) and race-ethnicity (Non-Hispanic White, Non-Hispanic black, Hispanic, and other).

[b] Adjusted for variables in model 1 plus serum cotinine (log-transformed, continuous), time of venipuncture (morning, afternoon, and evening), season of collection (1 November through 30 April, 1 May through 31 October), ratio family income to poverty (continuous).

c Adjusted *p*-trend:*p*-values adjusted for multiple testing.

axis function, which would alter the hormonal milieu[37–40]. In the present investigation, we were unable to establish an association between blood lead or cadmium levels and serum TT levels among children and adolescent males, which suggested that current exposure levels in our population did not adversely affect testosterone synthesis and regulation.

To our best knowledge, this is the first study to look at the relationships between cadmium exposure and circulating TT levels in adolescent female subjects. The positive association between cadmium and TT levels in female adolescents found in this study was in agreement with the findings reported by Ali I et al. [16] and Nagata C et al. [19]. Ali I et al investigated the relationship between blood cadmium exposure and serum sex hormone levels in 438 postmenopausal Swedish women without hormone replacement therapy, a positive association between blood cadmium and serum TT levels, and an inverse association between blood cadmium and serum estradiol levels and the estradiol/testosterone ratio was found [16]. It is possible that cadmium with LH-induced P450 aromatase activity was responsible for the conversion of testosterone to estradiol. Das and Mukherjee showed that LH-stimulated P450

**Table 7. Percent differences (95% CI) in serum TT by quartiles of blood manganese exposure, NHANES, 2011–2012using weighted regression models.**

| Blood Manganese(µg/L) | Model 1[a] | Model2[b] |
|---|---|---|
| Male children(N = 431) | | |
| ≤7.86 (n = 108) | Reference | Reference |
| 7.86–9.72 (n = 108) | 18.17 (-9.85,54.91) | 16.09 (-10.90,51.24) |
| 9.72–11.84 (n = 108) | 47.31 (13.23,91.66) | 48.55 (14.80,92.22) |
| >11.84 (n = 107) | 12.24 (-14.41,47.20) | 10.72 (-15.09,44.38) |
| Adjusted *p*-trend | 0.40 | 0.46 |
| Male adolescents(N = 493) | | |
| ≤7.62 (n = 124) | Reference | Reference |
| 7.62–9.33 (n = 124) | 38.64 (11.7,72.06) | 31.51 (6.78,61.96) |
| 9.33–11.54 (n = 123) | 25.2 (0.38,56.16) | 15.49 (-6.81,43.13) |
| >11.54 (n = 122) | 18.39 (-5.89,48.94) | 11.88 (-10.38,39.68) |
| Adjusted *p*-trend | 0.40 | 0.60 |
| Female children(N = 426) | | |
| ≤8.72 (n = 107) | Reference | Reference |
| 8.72–10.52 (n = 107) | -19.44 (-33.33,-2.65) | -21.3 (-34.91,-4.85) |
| 10.52–12.90 (n = 107) | -4.94 (-21.88,15.68) | -6.5 (-23.13,13.72) |
| >12.90 (n = 105) | -1.51 (-19.36,20.29) | -3.71 (-21.25,17.72) |
| Adjusted *p*-trend | 0.49 | 0.60 |
| Female adolescents(N = 470) | | |
| ≤8.64 (n = 119) | Reference | Reference |
| 8.64–10.84 (n = 116) | 23.04 (8.64,39.36) | 22.53 (7.98,39.04) |
| 10.84–13.26 (n = 118) | 34.59 (19.41,51.71) | 36.26 (20.72,53.81) |
| >13.26 (n = 117) | 24.96 (10.47,41.36) | 24.81 (10.31,41.23) |
| Adjusted *p*-trend[c] | 0.001 | 0.0008 |

[a] Adjusted for age (continuous), BMI (normal/underweight, overweight, and obese) and race-ethnicity (Non-Hispanic White, Non-Hispanic black, Hispanic, and other).

[b] Adjusted for variables in model 1 plus serum cotinine (log-transformed, continuous), time of venipuncture (morning, afternoon, and evening), season of collection (1 November through 30 April, 1 May through 31 October), ratio family income to poverty (continuous).

c Adjusted *p*-trend:*p*-values adjusted for multiple testing.

aromatase activity and P450arom gene expression in carp ovarian follicles were significantly inhibited by cadmium chloride ($CdCl_2$) [41].Unfortunately, gonadotropins and estradiol were not measured in the 2011–2012 NHANES survey. Nagata C et al. found a significantly positive relationship between urinary cadmium and serum TT levels in 164 postmenopausal Japanese women [19]. In a study on rats, exposure to either moderate or high doses of dietary cadmium during gestation led to an increase in serum TT levels in the female rat offspring[42].

We observed a significant positive association between blood manganese and serum TT levels in female adolescents, whereas no association between blood manganese and serum TT levels was found in male adolescents. Several animal studies as reviewed by Dees [43], showed that exposing female and male rat pups to low but increased manganese levels caused the release of hypothalamic luteinizing hormone-releasing hormone (LHRH) in the serum of animals. This effect of manganese on the LHRH-releasing system was consistent with the elevated serum LH, FSH, and gonadal steroid levels in both sexes. Interestingly, sex differences were also observed; males required a higher manganese dose to influence the hypothalamic system

compared with females, which indicated that male hypothalamic systems were less sensitive to the influences of manganese. The reason for these sex differences might be due to diversities in the manganese metabolism between the two sexes since male rats can clear the metal over twice faster than females [44].

We also observed a significant positive association between blood selenium and serum TT levels in male adolescents. Selenium is an essential trace element required primarily for the maintenance of spermatogenesis and male fertility [45]. In previous studies, appropriate selenium levels appeared to exert a positive influence on Leydig cells, which thus influenced the secretion of testosterone [46]. An epidemiologic study also revealed that serum TT tended to increase in infertile men supplemented with selenium [47].

Certain limitations of the present study should be acknowledged. First, NHANES is a survey with a cross-sectional design, which restricts the proper interpretation of the causal associations between metal exposures and serum TT levels. Second, no data on free testosterone, SHBG, or other hormones or markers that might have provided clues regarding the mechanisms and/or sites of action of blood metals, were available. Third, the group of 12-to-19-year-old boys and girls classified as adolescents could have included a mixture of pre- and postpubescent children. Lastly, there might have been other confounding factors that we did not evaluate in our analysis.

## Conclusions

The results from this study suggest that the blood levels of certain metals were associated with altered serum TT concentrations in male and female adolescents, included in NHANES 2011–2012 data. Our findings pertain to low-level environmental metals exposure and might not be generalizable to environmental or occupational settings where higher metal doses could be encountered. Additionally, altered T levels were found to be linked to a wide range of adverse health effects, but additional epidemiologic human studies, as well as mechanistic studies, are needed to confirm the results of our analysis.

## Supporting information

**S1 File. Data name related to this study can be found in S1 File.**
(XLS)

**S2 File. Data related to this study can be found in S2 File.**
(XLS)

## Acknowledgments

We are grateful for the editors and reviewers.

## Author Contributions

**Data curation:** Ziwei Qian, Zijing Cai.

**Funding acquisition:** Ge Zhou, Luyao Zhang, Rongkui Hu.

**Investigation:** Qi Yao, Meilin Xu.

**Methodology:** Jianguo Dai.

**Project administration:** Rongkui Hu.

**Supervision:** Yong Tan.

**Writing – original draft:** Qi Yao.

**Writing – review & editing:** Ge Zhou, Meilin Xu, Jianguo Dai, Luyao Zhang, Yong Tan, Rongkui Hu.

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
