## [Decision Letter · Decision Letter 0]

15 Jul 2019

PONE-D-19-17206

Blood metal levels and serum testosterone concentrations in male and female children and adolescents: NHANES 2011–2012

PLOS ONE

Dear Dr. Hu,

Thank you for submitting your manuscript to PLOS ONE. After careful consideration, we feel that it has merit but does not fully meet PLOS ONE’s publication criteria as it currently stands. Therefore, we invite you to submit a revised version of the manuscript.

The reviewers have  done a great job with detailed comments and suggestions for this manuscript. The authors should fully address these issues in revised manuscript. 

We would appreciate receiving your revised manuscript by Aug 29 2019 11:59PM. To enhance the reproducibility of your results, we recommend that if applicable you deposit your laboratory protocols in protocols.io, where a protocol can be assigned its own identifier (DOI) such that it can be cited independently in the future. For instructions see: http://journals.plos.org/plosone/s/submission-guidelines#loc-laboratory-protocols

We look forward to receiving your revised manuscript.

Kind regards,

Yi Hu

Academic Editor

PLOS ONE

Journal Requirements:

1. Thank you for including your funding statement; "Yes"

Please provide an amended Funding Statement that declares *all* the funding or sources of support received during this specific study (whether external or internal to your organization) as detailed online in our guide for authors at http://journals.plos.org/plosone/s/submit-now.  

Please state what role the funders took in the study.  If any authors received a salary from any of your funders, please state which authors and which funder. If the funders had no role, please state: "The funders had no role in study design, data collection and analysis, decision to publish, or preparation of the manuscript."

Reviewers' comments:

Reviewer's Responses to Questions

**Comments to the Author**

1. Is the manuscript technically sound, and do the data support the conclusions?

Reviewer #1: No

Reviewer #2: Yes

Reviewer #3: Yes

Reviewer #4: Yes

Reviewer #5: Yes

2. Has the statistical analysis been performed appropriately and rigorously? 

Reviewer #1: No

Reviewer #2: Yes

Reviewer #3: Yes

Reviewer #4: Yes

Reviewer #5: Yes

3. Have the authors made all data underlying the findings in their manuscript fully available?

Reviewer #1: Yes

Reviewer #2: Yes

Reviewer #3: Yes

Reviewer #4: Yes

Reviewer #5: Yes

4. Is the manuscript presented in an intelligible fashion and written in standard English?

Reviewer #1: Yes

Reviewer #2: Yes

Reviewer #3: Yes

Reviewer #4: Yes

Reviewer #5: Yes

5. Review Comments to the Author

Reviewer #1: Dear Editor,

Thank you for the opportunity to provide a review of Manuscript PONE-D-19-17206, a research article entitled “Blood metal levels and serum testosterone concentrations in male and female children and adolescents: NHANES 2011–2012.” My comments relate primarily to the adequacy of the implementation and reporting of epidemiologic and statistical procedures.

The quality of the technical English was generally appropriate, but there are a number of errors in the text. The manuscript requires a thorough round of copyediting review. Nevertheless, the errors offered no bar to the clear assessment of the issues in the manuscript.

There are three major issues in this manuscript:

1. The authors need to justify their decision to ignore the use of sampling weights arising from the complex survey design of the NHANES. The justifications they provide -- accounting for the design effect is inefficient, that accounting for the oversampling in statistical models is appropriate, and that these technqiues have been used by others before -- is inadequate. First, the argument that others’ methods are applicable here is specious. There is no *positive* evidence that research is similar to the others. The danger of blindly following others is well known. Second, the inefficiency of the design effect and consideration of oversampling via statistical models stem from a serious misunderstanding of the paper by Korn and Graubard (appearing as reference 28 in the manuscript). I suggest that the authors study this paper carefully. It is my recommendation that the authors:

1. provide direct evidence that the use of their unweighted analysis meets the strict assumptions described by Korn and Graubard;

2. conduct weighted, unweighted and partially weighted analyses to determine the amount of inefficiency;

3. using the information from (a) and (b) above, provide clear and specific descriptions of the approach they will take.

I am not prepared to accept justification stemming from an erroneous understanding of the advice provided by Korn and Graubard that has then been adopted by other researchers as *carte blanche* permission to ignore the complex sampling design of the NHANES.

2. The authors have used a very crude method to approximate the level of heavy metal concentration when it is below the level of detection. They’re imputing a single value -- detection limit divided by the square root of two. This has the effect of reducing variability substantially, especially for cadmium. The detection limit is a hard limit. That is to say, replacing this limit with a single number only shifts the limit, but does not resolve the presence of a limit. For example, the limit for cadmium is reported by the authors to be 0.16 micrograms per litre. Table 2 shows that 72.2% of male children had levels below this limit. Thus, for these children, the limit was replaced with an imputed value of 0.16/sqrt(2) = 0.113. As I described, this simply shifted the original lower limit. The decision to do this means that the standard errors have been deflated, causing spurious reductions in the p-value. The authors need to consider more sophisticated techniques, possible through regression methods, that can resolve this. (In this setting, the data are termed “left truncated”.)

3. The authors have performed multiple testing on the same data and their results need to be considered quite carefully because it might be a chance finding. Let me demonstrate. The authors tested five metals (Cd, Pb, Hg, Mn, Se) on four groups (male children, female children, male adolescents, female adolescents) under three models (model 1 and model 2 and test for trend). Thus, the number of independent analyses conducted on the data is 5 x 4 x 3 = 60. That they have found about eight statistically significant results is hardly surprising, especially when they were not expecting these results in the first place. I strongly recommend that the authors adjust the familywise error rate.

The combined effect of the three issues above imply that the presence of biased inferences is quite likely. These three issues must be resolved with urgency. Until then, it is difficult to give credence to the results, discussion and conclusions of the paper, as these might change substantially.

Finally, to my mind, the more important issue here is not the degree to which heavy metals affect the level of testosterone, but whether these levels are associated with levels that are in the range for clinical action. The important *clinical* question is whether heavy metal concentrations are related to clinical testosterone deficiency in children and adolescents.

I am unwilling to recommend the acceptance of this manuscript until such time as these issues are considered.

Reviewer #2: All required questions have been answered and that all responses meet formating specifications.

Comments:

1. General comments: Define all abbreviations the first time they are used. Abbreviations defined in the abstract should be redefined in the body of the manuscript.

2. The abbreviation for serum TT levels is not consistent throughout the manuscript,

3. There are many details are needed, For an example on the introduction and discussion include reference to many studies for which the model studied (human, rats, mice) is not identified and the route of exposure is not provided.

4. In the “Introduction and Discussion” section There are repetition in many paragraphs. Please avoid redundancies

5. In the “Discussion” section the authors need to explain the significance of their findings and the novelty of their study and should substantiate their findings with already available relevant scientific information.

6. Discussion is too long , hard to follow and could be better organized.

7. In the “Discussion” section 1 st paragraph ( line7). Please correct levers to levels

8. Parts of the Discussion section are particularly hard to understand. Professional assistance is recommended.

9. In result caption section . Table 2. please add table legand blew the table

Reviewer #3: The manuscript describes the data taken on metals levels in blood and their association with testosterone levels in children and adolescents. The authors have successfully correlated two metals concentrations (Cd and Mn) with testosterone concentrations in the subjects, highlighting the fact that environmental exposure to heavy metals may lead to altered testosterone levels in growing children. Overall the manuscript describes interesting findings, which are important to publish.

However, I have few questions in mind while reading this manuscript for review purpose, which the authors need to answer before the manuscript is accepted for publication

Reviewer #4: General comment

The aim of this study is to investigate the association between blood levels of metals/elements and serum levels of testosterone in US children and adolescents. This study is new due to the research of association between metals and testosterone in children and adolescent is scarce. The methods align with the study aim and design is acceptable. The article is written in standard English. The conclusions are presented appropriately and are supported by the data.

However, the description of methods and result is not comprehensive. Revision is necessary before further consideration.

Specific comments

Introduction:

More description of previous epidemiologic studies regarding the correlation of metals such as lead, mercury, manganese, selenium and levels of androgen hormones is necessary.

It is also necessary to give clear description of total testosterone (TT). Were serum T or serum TT levels given in the cited epidemiological studies? In the manuscript, sometimes serum T is given while sometimes serum TT is given. Please make these term clear.

At page 3, 3rd paragraph, the cited animal studies of Cd and Hg are not related to androgen hormone levels, more relevant literature is needed.

Material and Methods: full name of NCHS, LOD should be given.

Page 5, 2.2 Serum TT: information of blood sampling should be given, were the blood samples for serum TT collected at the same time point as blood samples of metals?

Page 5, last two lines at the bottom: sentence “ Serum TT was log-transformed… was skewed left.” should be in the Statistical Analysis section.

Page 6, 2.4 Covariates: what is the reason to choose these confounding variables? Is the selection of confounders based on literature or Directed Acyclic Graph (DAG) or other methods?

Page 7, line 1-2: Please give the explanation for “we combined underweight and normal weight in one category”. What is the background to do so?

Page 7, line 7: please give a short description of serum cotinine measurement. Sentence “ serum cotinine was log-transformed” should be in the Statistical Analysis section.

Page 7, Statistical Analysis:

The statistical power of category variables is generally weaker than continuous variables. For the association analysis of blood levels of metals and serum TT concentrations, authors only performed the linear regression analysis using quartiles and tertiles of metals while not run analysis for continuous metal data. Please clarify it.

In the Table 3-7, Model 1 and Model 2 are given. However, the description of Model 1 and Model 2 is not given here. Please clarify why use these two models and give the corresponding description in Statistical analysis section.

Results

Page 8, Line 2 -3 under Results: for sentence “ As expected, serum TT concentrations…. than male children.” Are the differences statistically significant? Are there significant differences of serum TT between female children and female adolescent? In addition, are there difference between genders and between children and adolescent for other characteristics parameters?

Page 8, line 3 from bottom: sentence “ A large portion (98.6%-84.0%) of the samples had blood lead and total mercury levels > LOD” is not clear. This sentence should be reformulated to separately describe lead and mercury.

Page 8, last line: sentence “ The median concentrations of all blood metals were higher in adolescents than in children. Are the differences statistically significant?

Page 10, line 6-8: sentence “ The mean serum TT was significantly lower for all quartiles of blood selenium than that of the lowest quartile in all population subgroups.” is not precise. This is the case only for female children. Please check!

Page 10, line 1-2 of last paragraph: sentence “ As seen in Table 7…. in all population subgroups” is not precise. This is the case only for female adolescents. Check!

Discussion: beware of “serum T” and “serum TT”. Are they same term?

Page11, line 2-6 of 2nd paragraph: sentences “ In a previous study, Meeker et al…. which is in contrast to the results of the present study.” is not clear. Did Meeker measure serum TT or serum T? What is the difference between serum T and serum TT in these sentences? Serum T is free testosterone?

Page 12, line 8-13: for sentences “ Differences in cadmium or lead levels…. in children and adolescent males, respectively”, it is doubtful to compare the present study with the cited studies (reference 32, 33) which were for adults while the present study is for children and adolescent because the level of both metals and androgens are different for adults and children, adolescents.

Page 15, line 2-3: sentence “ Our findings… may not be generalizable to environmental .. “ is not clear. Reformulation is necessary.

Table 3-7: sample numbers of each quartile or tertile should be given in the tables.

Supporting information is not mentioned in the text. What is the aim by giving this information?

Reviewer #5: The manuscript entitled "Blood metal levels and serum testosterone concentrations in male and female children and adolescents: NHANES 2011-2012"aimed to investigate whether there is an association between exposure to heavy metals and testosterone levels in children and adolescents. The work was well conducted, presents a relevant sample population and the results were explored through a consistent statistical analysis. The results indicated that some blood metals were positively associated with serum TT levels in female and male adolescents. However, the introduction of the article does not value the importance of the study. Much of the introduction (page 4, line 3-17) contains elements that are also cited in the discussion. Alternatively, the authors should explore in the introduction what are the sources or means of exposure of each of the heavy metals studied, as well as problematize how this exposure can impact the health of this specific population. Thus, I suggest that the authors make a comprehensive review of the introduction of the manuscript. In addition, a general review of writing is recommended, as several spelling errors were found throughout the text.

6. PLOS authors have the option to publish the peer review history of their article (what does this mean?). If published, this will include your full peer review and any attached files.

Reviewer #1: No

Reviewer #2: No

Reviewer #3: Yes: Dr. Tariq Mahmood

Reviewer #4: No

Reviewer #5: Yes: Fernanda Cristina Alcantara dos Santos

---

## [Author Response · Author response to Decision Letter 0]

23 Sep 2019

Dear editor:

Thank you very much for your letter and the reviewers' comments. We have carefully studied the comments of the reviewers and made corrections accordingly, which are highlighted in red in the revised version of the manuscript. Point-by-point responses are listed below. We hope that revised version is acceptable for publication in your journal. I am looking forward to hearing from you.

Sincerely Yours,

Rongkui Hu, M.M. 

Department of Reproductive Medicine, Affiliated Hospital of Nanjing University of Chinese Medicine, Jiangsu Province Hospital of Chinese Medicine, 155 Hanzhong Road, Nanjing, 210046, Jiangsu Province, China. 

E-mail: xiangyu198110@163.com

Responses to the reviewers 

First of all, we sincerely thank all reviewers for their positive and constructive comments and suggestions, which not only help improve our manuscript, but also provide some ideas for our future studies. We have carefully studied the comments of the reviewers and made corrections accordingly, which are highlighted in red in the revised version of the manuscript. Point-by-point responses are listed below.

Reviewer #1:

 Response to comment: The authors need to justify their decision to ignore the use of sampling weights arising from the complex survey design of the NHANES. The justifications they provide -- accounting for the design effect is inefficient, that accounting for the oversampling in statistical models is appropriate, and that these technqiues have been used by others before -- is inadequate. First, the argument that others’ methods are applicable here is specious. There is no *positive* evidence that research is similar to the others. The danger of blindly following others is well known. Second, the inefficiency of the design effect and consideration of oversampling via statistical models stem from a serious misunderstanding of the paper by Korn and Graubard (appearing as reference 28 in the manuscript). I suggest that the authors study this paper carefully. It is my recommendation that the authors:

1. provide direct evidence that the use of their unweighted analysis meets the strict assumptions described by Korn and Graubard;

2. conduct weighted, unweighted and partially weighted analyses to determine the amount of inefficiency;

3. using the information from (a) and (b) above, provide clear and specific descriptions of the approach they will take.

I am not prepared to accept justification stemming from an erroneous understanding of the advice provided by Korn and Graubard that has then been adopted by other researchers as *carte blanche* permission to ignore the complex sampling design of the NHANES.

Response: We appreciate the reviewer’s thoughtful comments. This is absolutely right. Sampling weights are critical in this complex survey design of NHANES. We rechecked the data and literatures and decided to use weighted analyses as the reviewer suggested. Details of the approach were described in the ‘Statistical Analysis’ section. The tables, results and conclusions were updated accordingly. We thank the reviewer again for this constructive comment, which helped us improve the quality of the manuscript. 

 Response to comment: The authors have used a very crude method to approximate the level of heavy metal concentration when it is below the level of detection. They’re imputing a single value -- detection limit divided by the square root of two. This has the effect of reducing variability substantially, especially for cadmium. The detection limit is a hard limit. That is to say, replacing this limit with a single number only shifts the limit, but does not resolve the presence of a limit. For example, the limit for cadmium is reported by the authors to be 0.16 micrograms per litre. Table 2 shows that 72.2% of male children had levels below this limit. Thus, for these children, the limit was replaced with an imputed value of 0.16/sqrt(2) = 0.113. As I described, this simply shifted the original lower limit. The decision to do this means that the standard errors have been deflated, causing spurious reductions in the p-value. The authors need to consider more sophisticated techniques, possible through regression methods, that can resolve this. (In this setting, the data are termed “left truncated”.)

Response: We thank the reviewer for this valuable comment. We agree that replacing the limit with a single number does not solve the problem of measuring limitation. In the meantime, we understand that truncated regression is commonly used when sample has been truncated. However, the truncated variable should be dependent variable. In our study, the dependent variable is the serum total testosterone that was not truncated. The truncated variables here are blood metal concentrations. Replacing the limit with a single number does not help for detection limit, but here we did not use the observed values. Instead, we used the categories described in the ‘Statistical Analysis’ section, e.g. using quartiles when more than 75% of the samples were above the limit. In this case, it does not matter whether we replaced the unobserved values with a single number because it does not change the rank/order of those subjects. Thus it does not affect the categories and the results will not change. We appreciate the reviewer’s thoughtful comment, but since this is beyond the scope of this paper, we did not include truncated regression in the manuscript. 

 Response to comment: The authors have performed multiple testing on the same data and their results need to be considered quite carefully because it might be a chance finding. Let me demonstrate. The authors tested five metals (Cd, Pb, Hg, Mn, Se) on four groups (male children, female children, male adolescents, female adolescents) under three models (model 1 and model 2 and test for trend). Thus, the number of independent analyses conducted on the data is 5 x 4 x 3 = 60. That they have found about eight statistically significant results is hardly surprising, especially when they were not expecting these results in the first place. I strongly recommend that the authors adjust the familywise error rate.

Response: We thank the reviewer for pointing this out. We agree multiple testing should be considered here. Model 2 was used to verify the findings in model 1, thus we did not consider this in multiple testing. Also, the five metals were fairly independent; we did not consider it in multiple testing either. However, we did perform multiple testing for the four subgroups (male children, female children, male adolescents, female adolescents). The p-values were adjusted for multiple testing using least significant difference (LSD) method. Detailed modification can be found in the ‘Statistical Analysis’ section. All p-valued were updated with adjusted p-values.

 Response to comment: Finally, to my mind, the more important issue here is not the degree to which heavy metals affect the level of testosterone, but whether these levels are associated with levels that are in the range for clinical action. The important *clinical* question is whether heavy metal concentrations are related to clinical testosterone deficiency in children and adolescents.

Response: We are genuinely grateful and appreciative for the reviewer’s thorough comments and suggestion to improve the quality of the manuscript. The authors carefully addressed the three issues arisen by the reviewer. Changes and modifications were made accordingly. In this study, we are trying to detect the difference in level of testosterone between subjects with high and low metal exposures. This will provide some clinical guidance in polycystic ovary syndrome (PCOS), cancer, altered pubertal development, hormone imbalances et al. An estimated 5 to 7 million women in the United States (U.S) suffer with the effects of PCOS; and PCOS can occur in girls as young as 11 years old. PCOS is the most common hormonal disorder among women of reproductive age and is the leading cause of infertility. Altered pubertal development may be at greater risk for psychological disorders and alcohol and substance abuse during adolescence. Although clinical testosterone deficiency in children and adolescents is an important problem, it is not our primary aim of this study. We are considering this in our next project. Again, we appreciate the reviewer’s comments, which were of great help to improve our work. 

Reviewer #2:

 Response to comment: General comments: Define all abbreviations the first time they are used. Abbreviations defined in the abstract should be redefined in the body of the manuscript.

Response: We are very sorry for our negligence. All abbreviations for the first time have been defined in the body of the manuscript.

 Response to comment: The abbreviation for serum TT levels is not consistent throughout the manuscript.

Response: We are sorry for our negligence. We have corrected it.

 Response to comment: There are many details are needed, For an example on the introduction and discussion include reference to many studies for which the model studied (human, rats, mice) is not identified and the route of exposure is not provided.

In the “Introduction and Discussion” section, there are repetition in many paragraphs. Please avoid redundancies.

In the “Discussion” section the authors need to explain the significance of their findings and the novelty of their study and should substantiate their findings with already available relevant scientific information.

Discussion is too long, hard to follow and could be better organized.

Response: We are genuinely grateful for the reviewer’s suggestion to improve the quality of the manuscript. We have reorganized the “Introduction and Discussion” section according to your comments. Please see the “Introduction and Discussion” section.

 Response to comment: Parts of the Discussion section are particularly hard to understand. Professional assistance is recommended.

Response: According to your comment, this manuscript has been carefully re-checked by us and then it was sent to a professional English editing company for language correction.

 Response to comment: In result caption section. Table 2. please add table legand blew the table

Response: We thank the reviewer for pointing this out. We have added table legand below the table 2. Please see Table2.

Reviewer #3:

Response to comment: The manuscript describes the data taken on metals levels in blood and their association with testosterone levels in children and adolescents. The authors have successfully correlated two metals concentrations (Cd and Mn) with testosterone concentrations in the subjects, highlighting the fact that environmental exposure to heavy metals may lead to altered testosterone levels in growing children. Overall the manuscript describes interesting findings, which are important to publish.

However, I have few questions in mind while reading this manuscript for review purpose, which the authors need to answer before the manuscript is accepted for publication.

Response: We are genuinely grateful and appreciative for the reviewer’s thorough comments to improve the quality of the manuscript. Most studies assessing the reproductive endocrine-disrupting effect of low, environmentally-relevant metal exposures in adults have focused on non-essential metals (cadmium and lead), whereas information on other potential endocrine-disrupting metals is still scant. Moreover, children and adolescents are at the ages where susceptibility to the adverse health effects of endocrine disruptors is most concerning. However, few studies have focused on the relationship between environmental metal exposure and endogenous androgen hormone levels in children and adolescents. Therefore, in this study, we are trying to detect the difference in level of testosterone between subjects with high and low environmental metals exposure (cadmium, lead, mercury, manganese, and selenium) in children and adolescents. This will provide some clinical guidance in polycystic ovary syndrome (PCOS), cancer, altered pubertal development, hormone imbalances et al. An estimated 5 to 7 million women in the United States (U.S) suffer with the effects of PCOS; and PCOS can occur in girls as young as 11 years old. PCOS is the most common hormonal disorder among women of reproductive age and is the leading cause of infertility. Altered pubertal development may be at greater risk for psychological disorders and alcohol and substance abuse during adolescence. 

Reviewer #4: 

Introduction:

 Response to comment: More description of previous epidemiologic studies regarding the correlation of metals such as lead, mercury, manganese, selenium and levels of androgen hormones is necessary.

Response: We are genuinely grateful the reviewer’s suggestions to improve the quality of the manuscript. We have reorganized the “Introduction” section according to your suggestions. Please see paragraph 3 of the “Introduction” section in page 3-4.

 Response to comment: It is also necessary to give clear description of total testosterone (TT). Were serum T or serum TT levels given in the cited epidemiological studies? In the manuscript, sometimes serum T is given while sometimes serum TT is given. Please make these term clear.

Response: We are very sorry for our negligence. We have made these terms clear (testosterone and serum total testosterone). We also have explained the differences between testosterone and serum total testosterone. Please see paragraph 2 of the “Discussion” section in page 11-12.

 Response to comment: At page 3, 3rd paragraph, the cited animal studies of Cd and Hg are not related to androgen hormone levels, more relevant literature is needed.

Response: We are genuinely grateful and appreciative for the reviewer’s suggestions to improve the quality of the manuscript. We have reorganized the “Introduction” section according to your suggestions. Please see paragraph 3 of the “Introduction” section in page 3-4.

Material and Methods:

 Response to comment: full name of NCHS, LOD should be given.

Response: We are very sorry for our negligence. Full name of NCHS, LOD have been given. Please see paragraph 1of page 5 and paragraph 2 of page 6.

 Response to comment: Page 5, 2.2 Serum TT: information of blood sampling should be given, were the blood samples for serum TT collected at the same time point as blood samples of metals?

Response: We are sorry for our negligence. Information of blood sampling have been given, please see section 2.2 in page 5.The blood samples for serum TT was collected at the same time point as blood samples of metals.

 Response to comment: Page 5, last two lines at the bottom: sentence “Serum TT was log-transformed… was skewed left.” should be in the Statistical Analysis section.

Response: We have accepted your suggestion, please see the “Statistical Analysis”section.

 Response to comment: Page 6, 2.4 Covariates: what is the reason to choose these confounding variables? Is the selection of confounders based on literature or Directed Acyclic Graph (DAG) or other methods?

Response: We choosed the following as potential confounding variables: age, race/ethnicity, poverty income ratio (PIR), obesity, seasons of collection, times of venipuncture, and serum cotinine. Age and race/ethnicity are demographic factors. PIR is a socioeconomic factor. Obesity is associated with androgen imbalances in body（Escobar-Morreale HF, Santacruz E, Luque-Ramírez M, Botella Carretero JI. Prevalence of 'obesity-associated gonadal dysfunction' in severely obese men and women and its resolution after bariatric surgery: a systematic review and meta-analysis. Hum Reprod Update. 2017; 23(4):390-408.）. Levels of serum TT in body have a wide variation due to diurnal, weekly and seasonal variations. The diurnal variations lead to a peak in the serum T levels in the early morning followed by a progressive decline to the nadir in the evening. Nadir values are approximately 15% lower than the peak morning values (Paduch DA, Brannigan RE, Fuchs EF, Kim ED, Marmar JL, Sandlow JI. The laboratory diagnosis of testosterone deficiency. Urology. 2014; 83(5): 980-988.). Serum cotinine is a biomarker of exposure to environmental tobacco smoke. The selection of these confounders was based on literatures (such as 1. Meeker JD, Rossano MG, Protas B, Padmanahban V, Diamond MP, Puscheck E, Daly D, Paneth N, Wirth JJ. Environmental exposure to metals and male reproductive hormones: circulating testosterone is inversely associated with blood molybdenum. Fertil Steril. 2010; 93(1): 130-140. 2. Menke A, Guallar E, Shiels MS, Rohrmann S, Basaria S, Rifai N, Nelson WG, Platz EA. The association of urinary cadmium with sex steroid hormone concentrations in a general population sample of us adult men. BMC Public Health 2008; 8, 72. 3. Scinicariello F, Buser MC. Serum Testosterone Concentrations and Urinary Bisphenol A, Benzophenone-3, Triclosan, and Paraben Levels in Male and Female Children and Adolescents: NHANES 2011-2012. Environ Health Perspect. 2016; 124(12):1898-1904.).

 Response to comment: Page 7, line 1-2: Please give the explanation for “we combined underweight and normal weight in one category”. What is the background to do so?

Response: We thank the reviewer for the comment. There are two reasons that we combined underweight and normal weight in one category. First, obesity is associated with associated with androgen imbalances in body. （Escobar-Morreale HF, Santacruz E, Luque-Ramírez M, Botella Carretero JI. Prevalence of 'obesity-associated gonadal dysfunction' in severely obese men and women and its resolution after bariatric surgery: a systematic review and meta-analysis. Hum Reprod Update. 2017; 23(4):390-408.）Second, we combined underweight and normal weight in one category, which are based on literatures (1. Scinicariello F, Buser MC. Serum Testosterone Concentrations and Urinary Bisphenol A, Benzophenone-3, Triclosan, and Paraben Levels in Male and Female Children and Adolescents: NHANES 2011-2012. Environ Health Perspect. 2016; 124(12):1898-1904. 2. Kresovich, J. K., Argos, M. & Turyk, M. E. Associations of lead and cadmium with sex hormones in adult males. Environ. Res. 2015; 142: 25–33. ).

 Response to comment: Page 7, line 7: please give a short description of serum cotinine measurement. Sentence “ serum cotinine was log-transformed” should be in the Statistical Analysis section.

Response: We have accepted your suggestion, please see the “Statistical Analysis”section.

Statistical Analysis:

 Response to comment: Page 7, The statistical power of category variables is generally weaker than continuous variables. For the association analysis of blood levels of metals and serum TT concentrations, authors only performed the linear regression analysis using quartiles and tertiles of metals while not run analysis for continuous metal data. Please clarify it.

Response: We thank the reviewer for this valuable comment. Using quartiles and tertiles for continuous variable will lose some power. However, there are two reasons that we use quartiles and tertiles, not the continuous values. First, there are significant amount of blood levels of metals were unobservable due to the lower detection limit (LOD). A commonly used method is to replace all unobservable values below LOD by a single value, e.g. LOD/2 or LOD/√2. But this does not solve the problem, it just shifts the limits. If we use quartiles and tertiles, the results will not be affected. It does not matter whether we replaced the unobserved values with a single number because it does not change the rank/order of those subjects. Thus it does not affect the categories and the results will not change. Second, we are based on literatures, (1. Scinicariello F, Buser MC. Serum Testosterone Concentrations and Urinary Bisphenol A, Benzophenone-3, Triclosan, and Paraben Levels in Male and Female Children and Adolescents: NHANES 2011-2012. Environ Health Perspect. 2016; 124(12):1898-1904. 2. Kresovich, J. K., Argos, M. & Turyk, M. E. Associations of lead and cadmium with sex hormones in adult males. Environ. Res. 2015;142: 25–33. ) which used quantiles and tetrtiles most of the time. 

 Response to comment: In the Table 3-7, Model 1 and Model 2 are given. However, the description of Model 1 and Model 2 is not given here. Please clarify why use these two models and give the corresponding description in Statistical analysis section.

Response: We thank the reviewer for the comment. We have descripted in “Statistical analysis” section, please see page 8 line6-8, Model 1 controlled for age, race and BMI. Model 2 controlled for PIR, seasons of collection, times of venipuncture, and serum cotinine, in addition to the covariates of model 1. Model Model 1 is only to adjust demographic factors. In addition to adjusting the demographic factors, model 2 also adjusted for confounding factors that may affect serum TT levels in our sample. These confounding factors were based on literatures. (such as Scinicariello F, Buser MC. Serum Testosterone Concentrations and Urinary Bisphenol A, Benzophenone-3, Triclosan, and Paraben Levels in Male and Female Children and Adolescents: NHANES 2011-2012. Environ Health Perspect. 2016; 124(12):1898-1904. Meeker JD, Rossano MG, Protas B, Padmanahban V, Diamond MP, Puscheck E, Daly D, Paneth N, Wirth JJ. Environmental exposure to metals and male reproductive hormones: circulating testosterone is inversely associated with blood molybdenum. Fertil Steril. 2010; 93(1): 130-140.)

Results

 Response to comment:Page 8, line 3 from bottom: sentence “A large portion (98.6%-84.0%) of the samples had blood lead and total mercury levels > LOD” is not clear. This sentence should be reformulated to separately describe lead and mercury.

Response: We have accepted your suggestion, please see paragraph 2 of the “Result”section in page 8-9. 

 Response to comment:Page 8, last line: sentence “The median concentrations of all blood metals were higher in adolescents than in children. Are the differences statistically significant?

Response: We are very sorry for our negligence. We have corrected it.

 Response to comment:Page 10, line 6-8: sentence “ The mean serum TT was significantly lower for all quartiles of blood selenium than that of the lowest quartile in all population subgroups.” is not precise. This is the case only for female children. Please check!

Page 10, line 1-2 of last paragraph: sentence “ As seen in Table 7…. in all population subgroups” is not precise. This is the case only for female adolescents. Check!

Response: We thank the reviewer for the comment. According to reviewer#1 suggestions, we employed weighted multivariable linear regression models using NHANES sampling weights to evaluate the association between log-transformed serum TT. p-values were adjusted for multiple testing using least significant difference (LSD) method. Details of the approach were described in the ‘Statistical Analysis’ section. The tables, results and conclusions were updated accordingly. 

Discussion: 

 Response to comment: beware of “serum T” and “serum TT”. Are they same term?

Response: We are very sorry for our negligence. We have corrected it.

16. Response to comment: Page11, line 2-6 of 2nd paragraph: sentences “ In a previous study, Meeker et al…. which is in contrast to the results of the present study.” is not clear. Did Meeker measure serum TT or serum T? What is the difference between serum T and serum TT in these sentences? Serum T is free testosterone?

Page 12, line 8-13: for sentences “ Differences in cadmium or lead levels…. in children and adolescent males, respectively”, it is doubtful to compare the present study with the cited studies (reference 32, 33) which were for adults while the present study is for children and adolescent because the level of both metals and androgens are different for adults and children, adolescents.

Response: We are genuinely grateful for the reviewer’s suggestion to improve the quality of the manuscript. We have reorganized the “Discussion” section according to your comments. Please see the “Discussion” section.

 Response to comment: Page 15, line 2-3: sentence “ Our findings… may not be generalizable to environmental .. “ is not clear. Reformulation is necessary.

Response: We are genuinely grateful for the reviewer’s comments to improve the quality of the manuscript. We have reformulated the sentence. Please see page 15, line 2-4.

 Response to comment: Table 3-7: sample numbers of each quartile or tertile should be given in the tables.

Response: We have shown sample numbers of each quartile or tertile in table3-7. Please see table 3-7.

Reviewer #5: 

Response to comment: The manuscript entitled "Blood metal levels and serum testosterone concentrations in male and female children and adolescents: NHANES 2011-2012"aimed to investigate whether there is an association between exposure to heavy metals and testosterone levels in children and adolescents. The work was well conducted, presents a relevant sample population and the results were explored through a consistent statistical analysis. The results indicated that some blood metals were positively associated with serum TT levels in female and male adolescents. However, the introduction of the article does not value the importance of the study. Much of the introduction (page 4, line 3-17) contains elements that are also cited in the discussion. Alternatively, the authors should explore in the introduction what are the sources or means of exposure of each of the heavy metals studied, as well as problematize how this exposure can impact the health of this specific population. Thus, I suggest that the authors make a comprehensive review of the introduction of the manuscript. In addition, a general review of writing is recommended, as several spelling errors were found throughout the text.

Response: We are genuinely grateful and appreciative for the reviewer’s thorough comments to improve the quality of the manuscript. We have reorganized the “Introduction and Discussion” section according to your comments. This manuscript has been carefully re-checked by us and then it was sent to a professional English editing company for language correction.

---

## [Decision Letter · Decision Letter 1]

11 Oct 2019

PONE-D-19-17206R1

Blood metal levels and serum testosterone concentrations in male and female children and adolescents: NHANES 2011–2012

PLOS ONE

Dear Dr. Hu,

Thank you for submitting your manuscript to PLOS ONE. After careful consideration, we feel that it has merit but does not fully meet PLOS ONE’s publication criteria as it currently stands. Therefore, we invite you to submit a revised version of the manuscript that addresses the points raised during the review process.

We would appreciate receiving your revised manuscript by Nov 25 2019 11:59PM. To enhance the reproducibility of your results, we recommend that if applicable you deposit your laboratory protocols in protocols.io, where a protocol can be assigned its own identifier (DOI) such that it can be cited independently in the future. For instructions see: http://journals.plos.org/plosone/s/submission-guidelines#loc-laboratory-protocols

We look forward to receiving your revised manuscript.

Kind regards,

Yi Hu

Academic Editor

PLOS ONE

Reviewers' comments:

Reviewer's Responses to Questions

**Comments to the Author**

1. If the authors have adequately addressed your comments raised in a previous round of review and you feel that this manuscript is now acceptable for publication, you may indicate that here to bypass the “Comments to the Author” section, enter your conflict of interest statement in the “Confidential to Editor” section, and submit your "Accept" recommendation.

Reviewer #1: All comments have been addressed

Reviewer #2: All comments have been addressed

Reviewer #3: All comments have been addressed

Reviewer #4: All comments have been addressed

Reviewer #5: All comments have been addressed

2. Is the manuscript technically sound, and do the data support the conclusions?

Reviewer #1: (No Response)

Reviewer #2: Yes

Reviewer #3: Partly

Reviewer #4: Yes

Reviewer #5: Yes

3. Has the statistical analysis been performed appropriately and rigorously? 

Reviewer #1: (No Response)

Reviewer #2: Yes

Reviewer #3: Yes

Reviewer #4: Yes

Reviewer #5: Yes

4. Have the authors made all data underlying the findings in their manuscript fully available?

Reviewer #1: (No Response)

Reviewer #2: Yes

Reviewer #3: Yes

Reviewer #4: Yes

Reviewer #5: Yes

5. Is the manuscript presented in an intelligible fashion and written in standard English?

Reviewer #1: (No Response)

Reviewer #2: Yes

Reviewer #3: Yes

Reviewer #4: Yes

Reviewer #5: Yes

6. Review Comments to the Author

Reviewer #1: (No Response)

Reviewer #2: After reviewing the manuscript that entitled (Blood metal levels and serum testosterone concentrations in male and female children and adolescents: NHANES 2011–2012) and make the first review and give some kind of modification . i find the following I accept all the modification and recommend to accept the manuscript

Reviewer #3: The revised version is in acceptable form now. can be proceeded further for publication in the PlosOne Journal

Reviewer #4: The author took into account of the raised questions and made the corresponding revision based on the comments. The revised manuscript is improved significantly.

A list of abbreviations can be given.

Page 6, section 2.5. Add” The selection of these confounders was based on literatures” in front of “ Race/ethnicity was… “

Reviewer #5: (No Response)

7. PLOS authors have the option to publish the peer review history of their article (what does this mean?). If published, this will include your full peer review and any attached files.

Reviewer #1: No

Reviewer #2: No

Reviewer #3: No

Reviewer #4: No

Reviewer #5: No

---

## [Author Response · Author response to Decision Letter 1]

22 Oct 2019

Dear editor:

Thank you very much for your letter and further instructions. We have carefully studied the comments and made corrections which we hope would meet with approval. Revisions are marked in red in the paper. The main corrections in the paper and responses to the reviewer’s comments are listed below. 

Sincerely yours,

Rongkui Hu, M.M. 

Department of Reproductive Medicine, Affiliated Hospital of Nanjing University of Chinese Medicine, Jiangsu Province Hospital of Chinese Medicine, 155 Hanzhong Road, Nanjing, 210046, Jiangsu Province, China. 

E-mail: xiangyu198110@163.com

Responses to the reviewers 

We sincerely thank all the reviewers for their positive and constructive comments and suggestions.

Reviewer #4:

1. Response to comment: Page 6, section 2.5. Add” The selection of these confounders was based on literatures” in front of “ Race/ethnicity was… ”

Response: We have accepted your good suggestion, Please see Page 6，section 2.5.

---

## [Editor Report · Decision Letter 2]

24 Oct 2019

Blood metal levels and serum testosterone concentrations in male and female children and adolescents: NHANES 2011–2012

PONE-D-19-17206R2

Dear Dr. Hu,

We are pleased to inform you that your manuscript has been judged scientifically suitable for publication and will be formally accepted for publication once it complies with all outstanding technical requirements.

With kind regards,

Yi Hu

Academic Editor

PLOS ONE
---

## [Editor Report · Acceptance letter]

29 Oct 2019

PONE-D-19-17206R2 

Blood metal levels and serum testosterone concentrations in male and female children and adolescents: NHANES 2011–2012 

Dear Dr. Hu:

I am pleased to inform you that your manuscript has been deemed suitable for publication in PLOS ONE. Congratulations! Your manuscript is now with our production department. 

With kind regards,

on behalf of

Prof. Yi Hu 

Academic Editor

PLOS ONE